# Inhibition of DNA Repair in Cancer Therapy: Toward a Multi-Target Approach

**DOI:** 10.3390/ijms21186684

**Published:** 2020-09-12

**Authors:** Samuele Lodovichi, Tiziana Cervelli, Achille Pellicioli, Alvaro Galli

**Affiliations:** 1Bioscience Department, University of Milan, Via Celoria 26, 20131 Milan, Italy; 87samuele@gmail.com; 2Yeast Genetics and Genomics Group, Laboratory of Functional Genetics and Genomics, Institute of Clinical Physiology CNR, Via Moruzzi 1, 56125 Pisa, Italy; tizicerv@ifc.cnr.it

**Keywords:** cell cycle checkpoint, DNA repair, cancer therapy, DNA repair inhibitors, synthetic lethality

## Abstract

Alterations in DNA repair pathways are one of the main drivers of cancer insurgence. Nevertheless, cancer cells are more susceptible to DNA damage than normal cells and they rely on specific functional repair pathways to survive. Thanks to advances in genome sequencing, we now have a better idea of which genes are mutated in specific cancers and this prompted the development of inhibitors targeting DNA repair players involved in pathways essential for cancer cells survival. Currently, the pivotal concept is that combining the inhibition of mechanisms on which cancer cells viability depends is the most promising way to treat tumorigenesis. Numerous inhibitors have been developed and for many of them, efficacy has been demonstrated either alone or in combination with chemo or radiotherapy. In this review, we will analyze the principal pathways involved in cell cycle checkpoint and DNA repair focusing on how their alterations could predispose to cancer, then we will explore the inhibitors developed or in development specifically targeting different proteins involved in each pathway, underscoring the rationale behind their usage and how their combination and/or exploitation as adjuvants to classic therapies could help in patients clinical outcome.

## 1. Introduction

Cells ability to faithfully repair DNA from insults from endogenous or exogenous sources is essential to maintain viability. Different kinds of damage arise from different sources and require specific DNA repair pathways, allowing recovery of the original sequence. Damage left unrepaired could be inherited after cell division, causing permanent genetic alteration. Accumulation of these mutations leads to cell senescence or apoptosis and may predispose to cancer development.

A complex network of sensors and effectors regulates DNA repair, which starts from the recognition of the damage on DNA to the choice of the most feasible repair pathway, depending on the type of damage and cell cycle phase. Among the various DNA repair pathways, we can distinguish: the base excision repair (BER) and the nucleotide excision repair (NER) primarily involved in removing small, non-helix-distorting base lesions from the genome and to repair bulky helix-distorting lesions and inter-strands crosslinks in G0/G1 phase, respectively [1,2]; the Fanconi anemia (FA) pathway acting during DNA replication for the repair of inter-strand crosslinks [3]; the mismatch repair (MMR) which is a system recognizing and repairing erroneous insertion, deletion, and mis-incorporation of bases [4]; the translesion DNA synthesis (TLS) which is a process that allows DNA replication machinery to replicate past DNA lesions by switching to specialized DNA polymerases [5]; the error-prone non-homologous end joining (NHEJ), which repairs DNA double strand breaks (DSBs) without the requirement of complementary strand [6]; the homologous recombination (HR) which faithfully repairs DSBs taking advantage of the sequence present on the complementary strand of the homologous chromosome or chromatid, acting in S and G2 cell cycle phases [7]. Moreover, DSBs can also be repaired by the annealing of short (1-few bp) or long (>100 bp) complementary sequences through the highly mutagenic MMEJ (microhomology-mediated end-joining, also known as alternative non-homologous end-joining or Alt-NHEJ), and single strand annealing (SSA), respectively [8,9]. Despite the specificity of each pathway for particular DNA damage and cell cycle phase, none of them are mutually exclusive and they form a network that involves proteins determining the repair outcomes. Mutations affecting the numerous players of these mechanisms could lead to an accumulation of unfaithfully repaired DNA, thus predisposing to cancer development. Moreover, alterations in processes that support genome integrity in normal cells, allow cancer cells to acquire aggressiveness and facilitate the emergence of resistance to DNA damaging cancer treatments [10]. Nevertheless, understanding the emerging role of DNA repair pathway deficiency in cancer allows the development of specific drugs targeting the remaining functional pathways, on which cancer cells are strongly dependent for survival and proliferation. This kind of approach is based on the concept of synthetic lethality, which is a genetic interaction between two genes. If the mutation occurs in only one of the two genes, cell viability is not affected; whereas when two genes are simultaneously mutated, cell death occurs [11]. The discovery of many synthetic lethal interactions between proteins involved in DNA repair allowed the development of personalized therapeutic treatments that target specific DNA repair enzymes to kill cancer cells.

Thanks to the advancement of next-generation sequencing technologies, we now have a clearer indication of which DNA repair-related genes are mutated in each specific cancer: for example, some breast and ovarian cancers are characterized by mutations in BRCA1/BRCA2, while alterations in MMR genes such as MSH2 or MLH1 are related to colorectal cancer [12]. Putting this information together with new synthetic lethality interactions, discovered by novel CRISPR/Cas9 screening approaches in human cells [13,14], will allow the development of the most effective therapeutic approaches by production of drugs targeting DNA repair enzymes specific for each pathological situation.

In this review, we will give a brief introduction to DNA damage checkpoints and repair focusing on different players involved in each pathway and describe how their alterations could impact cancer development and aggressiveness. Then, we will describe how these players can be targeted in cancer therapy and the most encouraging drugs under study, focusing on the rationale behind their usage to improve efficacy of classic therapy or induce specific synthetic lethal interactions.

## 2. DNA Damage Checkpoint Promotes DNA Damage Recognition and Cell Cycle Arrest: The First Barrier to Cancer Progression

When DNA damage arises, a complex network of signaling cascades named DNA damage checkpoint (DDC) are activated to recognize DNA damage and arrest the cell cycle, thus ensuring enough time to cells to repair the lesion or eventually leading to senescence/apoptosis. These processes are tightly regulated to guarantee cell survival by promoting genomic stability and reducing the possibility that lesions are inherited, after cell division. Similar to other cellular signaling cascades, DDC is driven by protein phosphorylation and at its core there are three upstream kinases: ATM, ATR, and DNA-PKs [15] (Figure 1); moreover, recent evidence also suggests that non-coding (nc)RNA molecules and the RNA interference (RNAi) factors Drosha and Dicer have a role in this mechanism [16,17].

Each kinase is recruited to DNA lesions by specific co-factors: ATM is recruited by the C terminus of NBS1 a component of the MRE11-RAD50-NBS1 (MRN) complex, DNA-PKs is recruited by Ku80/Ku70 heterodimer and ATR is recruited by its stable binding partner ATRIP-RPA. In general, the first two kinases are involved in the initial recognition of DSBs, then, ATM promotes either HR or NHEJ, while DNA-PK activity is restricted to NHEJ. Instead, ATR is recruited to all kind of lesions that generate ssDNA coated by replication protein A (RPA) such as: DSBs after end resection, bulky lesions when the damaged strand is excised by NER and stalled replication forks when the double helix is opened and/or processed by nucleases. ATR is also involved in the recognition of particular DNA damage such as UV-damaged DNA [18]. Following DNA damage recognition, all the three kinases determine cell cycle arrest by interacting with specific substrates (Figure 1). ATR and ATM phosphorylate and activate the tumor suppressor checkpoint kinase 1 (CHK1). CHK1 regulates the G2/M checkpoint by activating kinase WEE1, which in turn phosphorylates the Cyclin-dependent kinase 1 (CDK1) reducing its activity and preventing entry into mitosis; CHK1 also regulates S phase checkpoint by promoting CDC25A phosphatase degradation, whose activity is essential to remove inhibitory phosphate groups from kinases CDK4 and CDK2 and guarantee cell cycle progression [19].

Due to its activity among replication fork collapse, ATR-CHK1 axis protects cells from replication stress and mitotic catastrophe that could arise after uncontrolled division [20]. ATM and DNA-PK also phosphorylate the tumor suppressor checkpoint kinase 2 (CHK2); CHK2 stabilizes P53, which inhibits CDK6 and CDK4 leading to cell cycle arrest in G1/S phase [21].

Thanks to its inhibition of cell cycle progression, the activation of DDC is considered the first barrier to tumor progression and the loss of one or more DDC-involved proteins determines accumulation of un-repaired DNA lesions which greatly contribute to cancer initiation and progression. Indeed, DDC helps cells recover from DNA injuries and overexpression of its players is frequent in cancer resistant to DNA damaging agents.

ATM is a well-recognized tumor suppressor and its alterations are common in several cancers including breast, gastric, colorectal, and prostate cancer [22]. Mutations in DNA-PK have been found associated to several cancers such as prostate cancer and melanoma [23,24]. Finally, ATR mutations promote development of melanoma and oropharyngeal cancer syndrome [25,26].

After initial recognition of DNA damage and cell cycle arrest, cells undergo specific DNA repair pathway depending on the cell cycle phase and type of damage.

## 3. Defects in DSB Repair Greatly Promotes Cancer Development

DSBs are fatal DNA lesions that must be quickly repaired to prevent dangerous chromosomal rearrangements events. Several DSBs sensors must be located on the lesion to recognize it and activate downstream effectors. The initial proteins recruited to the DSBs are the Poly (ADP-ribose) polymerase 1 (PARP1) and Ku70/Ku80 complex: the first one catalyzes the formation and attachment of mono and/or poly (ADP-ribose) polymers to itself and target proteins, promoting chromatin decondensation and allowing binding of successive players; the second protein recognizes and binds to the DSBs ends recruiting DNA-PKs and NHEJ factors [27,28] (Figure 2).

Following this initial recognition, the MRN complex, composed of RAD50, NBS1, and MRE11, binds to the lesion and recruits ATM that promotes the checkpoint arrest. ATM also phosphorylates the histone H2AX, producing gamma-H2AX (γH2AX), which is essential for the recruitment of multiple factors, including the mediator of DNA damage checkpoint 1 (MDC1). This is accompanied by simultaneous accumulation of E3 ubiquitin-protein ligase RNF8, which further promotes chromatin relaxation allowing recruitment of additional DNA repair players [29].

At this point, the choice between HR and NHEJ is strongly dependent on cell cycle phase and resection process. In G1, DSB end resection is limited by several negative regulators, such as the Ku70/Ku80 heterodimer that is steadily bound to the broken ends of the DSB, the tumor suppressor p53-binding protein 1 (TP53BP1) that acts as a scaffold to regulate a network of proteins that prevent resection and several other factors that act as barriers [30]; activity of these proteins promote NHEJ [31,32].

Conversely, in S and G2 phases, HR is favored by DSB end resection stimulators mainly by CtBP- interacting protein (CtIP) in association with BRCA1 and MRN complex; CtIP recruitment relies on ATR and on the activity of E3 ubiquitin-protein ligase RNF138, which also promotes ubiquitination of Ku80 leading to its removal from DSB ends [33,34].

In HR (Figure 2A), the endonuclease activity of MRE11 nicks the strand several nucleotides away from the break and then resects the DNA towards the DSB; this “short-range” resection activity requires interaction with CtIP [35]. This process is thought to displace Ku70/Ku80 complex allowing access of “long range” resection factors, such as endonucleases DNA2 and EXO1, the Bloom syndrome helicase (BLM) and Werner syndrome ATP-dependent helicase (WRN) [36]. The ssDNA formed is rapidly coated by the replication protein A (RPA) that protects it from degradation; afterwards, RPA displacement is mediated by BRCA2 and PALB2 that also promote RAD51 binding to the ssDNA. Then, RAD51 initiates the search for homologous sequences and invasion of the complementary strand; this activity is stimulated by RAD52 as well, which also mediates the DNA-DNA interaction necessary for annealing of the complementary DNA strands [37,38,39]. After strand invasion, the replicative DNA polymerases (POL δ, ε) or translesion DNA polymerases (POL η, κ) extend the DNA strand generating a displacement loop (D-loop). At this point, D-loop structures can be solved by three pathways: double-strand break repair (DSBR), synthesis-dependent strand annealing (SDSA) or break induced replication (BIR); the first allowing the formation of crossover and non-crossover products, the second allowing only the formation of non-crossover and the third generating half-crossover products with loss of heterozygosis (LOH), frequently promoting mutagenesis [40]. It has been recently shown that TP53BP1 has also a role in regulating the last steps of HR by limiting helicases activity that prevents D-loop stability, therefore favoring crossover and BIR events [41].

In classic NHEJ (c-NHEJ) (Figure 2D), the binding of Ku70/Ku80 heterodimer to the DSB ends, together with TP53BP1, WRN helicase and several other barriers prevents end resection and recruits DNA-PK kinase. DNA-PK recruits the endonuclease ARTEMIS, which processes the broken ends until it finds cohesive nucleotides. In the last step, NHEJ factor 1 (XLF) interacts with DNA ligase 4 (LIG4) to catalyze the DSB ligation. Alternatively, DSBs can be repaired by SSA or MMEJ (Figure 2B,C) that, depending on the extension of end resection, results in DNA deletion. Both MMEJ and SSA also require ATM signaling [42] In SSA the resection process reveals flanking homologous sequences (>100 bp) that are annealed together by RAD52 and any gaps are filled by DNA polymerases; in MMEJ the Pol θ- associated helicase functions together with PARP1 to displace RPA from ssDNA, revealing short internal microhomologies (few bps) on the ssDNA ends and stabilizing their interaction. Then, the MRN complex recruits X-ray repair cross-complementing protein 1 (XRCC1) which form a complex with DNA ligase 3 (LIG3) to catalyze DNA ends ligation [43]. The choice between NHEJ and MMEJ is also dependent on WRN, which suppresses the recruitment of MRE11 and CtIP on the DSBs, thus promoting c-NHEJ [44]. Another critical step in MMEJ and SSA is the removal of the non-homologous 3′ ssDNA tail, which is mediated mainly by the XPF flap nucleases [45].

Considering the importance of DSB repair, alteration of the players is frequently related to cancer predisposition and development.

Accordingly, a comprehensive analysis across 33 cancer types identified HR pathway as the most frequently altered DNA repair pathway, particularly in ovarian cancer [46]. The most frequent mutated HR genes are *BRCA1* and *BRCA2*, followed by *RAD51*, *BLM,* and *RAD50* [46,47]. Germline mutations in *BRCA1* and *BRCA2* are related to the majority of hereditary breast and ovarian cancer (HBOC); however, beside these genes, several other low penetrance genes responsible of HBOC have been identified, such as *PALB2* and *RAD51* [48], and also mutations in genes coding for the components of the MRN complex [49,50].

Besides HBOC, mutations or alterations of HR-related genes are responsible for predisposition to other cancer types; for example, *RAD51* overexpression has been associated with poor prognosis in patients with solid malignancies [51]; mutations of *MRE11* are related to sporadic gastric cancer and neuroblastoma [52,53], while *RAD50* mutations are related to leukemia and endometrial carcinoma [54,55]; moreover, *NBS1* has emerged as a prostate and lung cancer-susceptibility gene [56,57].

Regarding cancer predisposition caused by alteration of c-NHEJ, TP53BP1 is the most interesting protein; when this protein is downregulated, it determines resistance to PARP inhibitors in breast and ovarian BRCA1 deficient cancer and to chemotherapeutic agents in colorectal cancer cells by reducing the protein level of the ATM-CHK2 pathway [58,59]. In general, by playing a pivotal role in the choice of DSB repair pathway, it has been extensively demonstrated that aberrant expression of the TP53BP1 protein contributes to tumor development [60].

Beside this, so far, few cancers are associated with the downregulation or alteration of genes involved in c-NHEJ. Only rare mutations of genes encoding for Ku70/Ku80, LIG4, ARTEMIS and XLF have been found in colon and endometrial cancer [24].

MMEJ and SSA are intrinsically mutagenic, generating deletions and causing genomic instability found in many human cancers. Initial DSB end resection is favored by CtIP, suggesting that this gene has oncogenic potential by promoting these pathways; accordingly, CtIP inactivation suppresses mammary tumorigenesis caused by p53 deficiency in mouse model [61]. Moreover, MMEJ relies on Pol-θ and elevated *POLQ* expression (encoding for Pol-θ) has been described in numerous cancer types, including breast and ovarian cancer [62].

Finally, having a role either in HR and NHEJ, alterations of DNA helicases such as: BLM, WRN, and REQLs predispose to tumorigenesis, in general their upregulation is involved in cell proliferation and resistance to DNA damaging agents, conversely their downregulation leads to genomic aberrations [63]. Particularly, *DNA2* alterations are related to genome instability by an enhanced end resection activity, caused by its overexpression in early stages of cancer [64]. Mutations of the *WRN* gene cause Werner syndrome, which is characterized by genetic instability and hematological disease [65], this helicase is often highly expressed in chronic myeloid leukemia determining increased cell survival through NHEJ [66].

## 4. Single Base Variations and Mismatched Base Pairs Are Repaired by Specific Pathways Linked to Cancer

BER is a pathway involved in repair of small base lesions such as oxidation, alkylation, and deamination (Figure 3A). The steps of the mechanism are: DNA damage recognition, excision of the base by a DNA glycosylase to generate an abasic (AP) site, and cleavage of the AP site by the AP endonuclease APE1 to form a DNA single strand break (SSB) [67]. When the SSB is formed, one of the first proteins activated is PARP1, which, acting as a SSB sensor, promotes recruitment of other enzymes involved in the repair process, such as XRCC1 [68,69]. Finally, the gap is filled by DNA polymerase β and the complex XRCC1 and LIG3 seals the nick, or as an alternative, two to 10 nucleotides are removed, a new nucleotide chain is synthetized by DNA polymerase and the final ligation step is performed by LIG1 (Figure 3A).

MMR is a pathway responsible for correcting mismatched base pairs and insertion/deletion loops (IDLs) that may occur during DNA replication [70] (Figure 3B). The central players are MutS proteins that form two heterodimers, MutSα (MSH2-MSH6) and MutSβ (MSH2-MSH3), and MutL proteins that form three heterodimers, MutLα (MLH1-PMS2), MutLβ (MLH1-MLH3), and MutLγ (MLH1-PMS1). MutS dimer recognizes the mismatched base on the daughter strand and binds the damaged DNA; MutL recruits the DNA Helicase II to separate the two strands. Then, the entire complex slides along the DNA, unwinding the strand that must be excised. Thereafter, the endonuclease activity of MLH1-PMS1, activated by the PCNA clamp produces a nick of the DNA strand [71]. Subsequently, the nicked DNA strand with the mismatch is excised by EXO1. Finally, POL δ synthesizes a new fragment and DNA ligase I (LIG1) catalyzes strand ligation [72].

There are few relations between BER and cancer; alterations in this pathway have been associated with colon and breast cancer [73,74]. Particularly, Pol-β expression seems to have a protective role against breast and lung carcinomas by impairing cancer cell metastasis due to an increased DNA de-methylation [75]; accordingly, Pol-β deficiency is associated with aggressive breast cancer [76]. Moreover, variants of this protein affecting its fidelity are associated to prostate and colon cancer [77,78] and could drive cellular transformation leading to cancer onset [79]. Instead, germline mutations in MMR genes are associated with hereditary non-polyposis colorectal cancer (HNPCC), also known as Lynch Syndrome, an autosomal dominant disease. Moreover, mutations in all the MMR genes, *MLH1* and *MSH2, PMS2* and *MSH6* have been found related to breast cancer susceptibility [50,80,81,82,83,84].

## 5. The DNA Crosslinks Repair Deals with NER and FA Pathways with Implication for Cancer

Inter-strand crosslinks (ICLs) covalently link the two strands of the DNA double helix and they arise from exposure to chemicals such as DNA damaging agents used in cancer therapy.

In quiescent cells (G0/G1), an ICL is recognized by NER pathway, which is also involved in the repair of intra-strand crosslinks and UV-damage [85] (Figure 4). In NER, two mechanisms of DNA damage sensing are known: one pathway recognizes damage all over the genome, and the other one recognizes damage in the transcribed strand of active genes that cause blockage of the RNA polymerase II. In the first case, detection is performed by the binding of the XPC complex (XPC, HR23B and CENT2) to the non-damaged strand, assisted by PARP1 activity [86]. In the second case, arrest of RNA polymerase II is the initial signal that promotes recruitment of the Cockayne syndrome group A (CSA) and group B (CSB) proteins. At this point, both pathways require XPA that acts as a scaffold organizing the other components of the pathway [87] and the transcriptional factor IIH complex (TFIIH) which first unwinds the DNA around the damage by the helicase activity of XPB and XPD. Then, DNA damage is removed by the excision repair protein ERCC-1-XPF and XPG endonuclease activity, generating a gap that is filled by DNA polymerase and by the DNA ligase activity of XRCC1-LIG3 or LIG1 [88].

In S phase, ICL lesions cause stalling of DNA replication forks that are recognized by the FA pathway (Figure 4). The anchor complex is the one recognizing the stalled replication fork; when activated, the anchor complex recruits a core complex which attaches a single ubiquitin to both FANCD2 and FANCI that together form the ID2 complex. This ubiquitination is essential for the activation of the nucleases involved in the unhooking of the DNA strands that is carried out by SLX4 and XPF [89], and also for the recruitment of the TLS polymerases that replace the stalled DNA polymerases and bypass the lesion via TLS; this pathway is promoted by ubiquitination of TLS regulator PCNA, which is performed by FANCD2 [90]. Among TLS polymerases, Pol ξ is the one involved in the lesion bypass step of ICL, while REV1 facilitates polymerase switching and coordinates extension steps [91]. FANCD2 deubiquitination is then performed by deubiquitinating complex USP1/UAF1. Finally, after DNA unhooking, the downstream players of the pathway coordinate HR to repair the DSB formed; these players are FANCS (BRCA1), FANCD1 (BRCA2), FANCN (PALB2), and FANCO (RAD51); they are referred to as downstream components of the FA pathway since their activation depends on the mono-ubiquitination step (Figure 4) [92].

Germline mutations of the FA-related genes determine development of FA which is a genetic disease causing an impaired response to DNA damage. The majority of the affected people develop cancer, most often leukemia [93]. FA disease can develop also by mutations of HR-related genes such as *BRCA1*, *BRCA2,* and *PALB2*; hence, the FA pathway is often called the FA/HR pathway. Alterations of FA-related proteins are also associated to breast and ovarian cancer development; mutations of FANCM, a component of the anchor complex, and FANCA, which is part of the core complex, are both correlated to increased risk of breast cancer [94].

## 6. Classic Anti-Cancer Therapy Relies on DNA Damage Induction to Which Cancer Cells Are More Sensitive

Cancer cells deficient in DNA damage checkpoint and repair pathways are particularly sensitive to DNA damage upon which chemo and radiation cancer therapies rely.

In chemotherapy, the most common agents used are alkylating agents, which cause mainly intra-strand and few inter-strand crosslinks by adding alkyl groups to DNA [95]. Currently, platinum-based chemotherapeutic drugs, called “alkylating-like”, are more frequently used because they form DNA crosslinks without having an alkyl group. Another group of DNA damaging agents used are inhibitors of topoisomerases, affecting the activity of two enzymes catalyzing the breaking and rejoining of the phosphodiester backbone of DNA strands during the normal cell cycle, topoisomerase I and topoisomerase II. Topoisomerase I inhibitors stabilize the cleavable complex of topoisomerases I, preventing DNA relegation and thus inducing DNA strand breaks; some examples are topotecan and camptothecin. Drugs against topoisomerase II consist of chemicals which target the complex DNA-topoisomerases II and promote cleavage activity or prevent re-ligation of the DNA, and inhibitors that reduce the turn-over of the enzyme; some examples are etoposide and doxorubicin [96].

A third group are cytotoxic antibiotics that act by different mechanisms of action and share the ability to interrupt cell division. Among this group, important examples are bleomycin and mitomycin C that damage DNA by producing free radicals and DNA inter-strand crosslinks, respectively [97].

Anti-metabolites are a group of molecules with a structure similar to nucleotides, but with altered chemical groups. These drugs exert their effect by either blocking the enzymes required for DNA synthesis or becoming incorporated into DNA and promoting DNA damage; some examples are capecitabine and gemcitabine [98].

Radiotherapy induces DNA damage using ionizing radiation (IR), either directly by ionizing the DNA strands or indirectly by producing free radicals as a result of ionized water molecules [99].

These treatments have proven to be effective against cancer, but resistance development is a common problem. Various factors drive drug resistance, such as increase of drug efflux, over-expression of DNA repair proteins or inhibition of proteins involved in apoptotic pathway [100]. To overcome this, it is now evident that combination with specific drugs targeting cancer deregulated pathways can greatly help, for this reason several compounds inhibiting different players in DDC and DNA repair are under study in pre-clinical and in clinical trials, alone or in combination with classic therapy.

## 7. Targeting DDC Is One of the Most Efficient Ways to Tackle Cancer Cells

Cancer cells accumulate DNA damage in an increased fashion as compared to normal cells and rely on DDC to avoid that excessive damage is inherited after cell division, for this reason several inhibitors targeting this pathway have been developed or are in development (Table 1).

As previously mentioned, DNA damage determines cell cycle arrest before entering mitosis, if this arrest is prolonged enough, cells go through the apoptotic pathway. Sometimes, cancer cells can avoid mitotic arrest and inappropriately enter mitosis before completion of the S and G2 phase; this frequently determines an improper distribution of chromosomes leading to mitotic catastrophe, which is an onco-suppressive signaling cascade preceding cell death via apoptosis, senescence or necrosis, in the first or subsequent cell divisions [101]. Inhibition of proteins that control the S/G2 checkpoint promotes this pathway and in solid cancer this is the main cell death-leading pathway induced by IR treatment [102,103].

ATM mutations could induce synthetic lethality in association with other mutations or specific drugs; this could be exploited to develop more accurate therapies [22]. Since ATM inhibition determines uncontrolled cell entry into mitosis by removing several checkpoints control, the use of ATM inhibitors in cancer therapy has been reported to cause sensitization to radiotherapy. In addition, ATM inhibition enhances cell sensitivity to topoisomerase I and II inhibitors such as camptothecin, etoposide, and doxorubicin [104]. Several ATM inhibitors have been used in cancer therapy. AZD0156 is a strong radio-sensitizer that recently has been demonstrated to inhibit tumor growth after radiation treatment of lung xenograft [105]; in addition, a combinatory effect with PARP inhibitors (olaparib) is under study since sensitivity to these drugs is related to ATM deficiency [106]. Currently, AZD0156 is in phase 1 trial (NCT02588105). A phase I clinical trial with the inhibitor AZD1390 (NCT03423628) (Table 1), which has been proved to radio-sensitize brain tumor models in pre-clinical studies, has also been initiated [107]. Several other compounds have been produced such as KU-55933, KU-60019 and KU-59403 that are in pre-clinical studies, therefore targeting ATM is a promising strategy for cancer treatment [108].

DNA-PK inhibition promotes radio-sensitivity in gastric cancer [109]; accordingly, its overexpression is associated to radio-resistance in thyroid, cervix, and prostate cancer [110] (Table 1). DNA-PKs deficiency can sensitize cells to DNA-damaging agents such as topoisomerase I and II inhibitors. This has led to the development of several DNA-PK inhibitors such as M3814, which has been demonstrated to increase efficacy of topoisomerase II inhibition in ovarian cancer models [111]. A clinical trial is ongoing to study efficacy of this drug in combination with radiotherapy and capecitabine, a drug that interferes with DNA and RNA metabolism (NCT03770689). Similarly, the DNA-PK inhibitor VX-984 enhances radio-sensitivity of glioblastoma and a clinical trial has been completed to assess safety of this drug in combination with chemotherapy (NCT02644278) [112]. Other drugs, currently under study, are CC-115 and MSC2490484A (NCT02516813) (NCT02833883), whose efficacy has been suggested in ATM-deficient cells and in combination with radiotherapy [113] (Table 1).

Differently from the other kinases, ATR does not participate in the initial recognition phases of DNA damage, but its activity is essential to prevent replication fork collapse, thus ensuring faithful duplication by binding to ssDNA and to arrest cell cycle in G2/M phase by interacting with CHK1 [114]. Since cancer cells are often defective in G1 phase checkpoint due to mutation of p53 and Rb tumor suppressor genes, they arrive in S phase with replication stress that could stall replication fork; therefore, cells rely on S-G2 phase checkpoint to avoid cell death due to excessive DSBs. Thus, a deficiency of ATR activity determines sensitivity to DNA damage agents such as IR, DNA cross-linking agents, and topoisomerase poisons [115]; its inhibition combined with PARP inhibitors also reduces tumor growth in BRCAness models [116] (Table 1). Currently, the ATR inhibitors M6620, AZD6738, BAY1895344, and VE-821 have been shown to increase lethality of cancer cell lines ATM, defective such as gastric cancer, non-small cell lung cancer and chronic lymphocytic leukemia, strongly suggesting a synthetic lethality interaction between the ATM and ATR pathways [117,118] (Table 1). A synthetic lethality interaction has been observed in triple-negative breast cancer cell lines also by combining AZD6738 with WEE1 inhibition, and with CHK1 inhibitor AZD7762, since combination of ATR and CHK1 inhibition leads to ssDNA accumulation and replication stress [119,120] (Table 1). In addition, since ATR inhibition enhances replication stress, as anticipated in pre-clinical trials, several studies analyze cancer cell sensitization by these drugs to DNA damaging agents such as cisplatin, gemcitabine, and radiation. In particular, M6620 was studied in combination with radiation in patients with brain metastases (NCT02589522) and with cisplatin for head and neck squamous cell carcinoma (NCT02567422) (Table 1); AZD6738 in combination with radiotherapy in patients with solid tumors, VE-821 in combination with radiation and chemotherapeutics in pancreatic cancer patients [121,122]. The effect of BAY1895344 in combination with chemotherapeutics or radiotherapy on tumor cell growth and survival was studied in cancer xenograft models that carry DNA damage repair deficiencies [123].

Targeting DDC pathway is an attractive strategy in cancer therapy. In fact, it has proven to be particularly effective in combination with radiotherapy determining cell death by induction of mitotic catastrophe pathway. Moreover, as DDC is upstream to several DNA repair pathways, different synthetic lethality combinations, currently under study, could be exploited taking advantage of this network.

## 8. Combination of DSB Inducers and DSB-Repair Proteins Inhibition Is Greatly Effective

A large number of DNA damaging agents exert their activity by inducing DSBs; therefore, combining these agents with compounds inhibiting DSB repair-involved players produced promising results. Particularly, inhibition of HR-involved proteins combined with DNA damage induction forces cells to use more error-prone DNA repair pathways, thus determining accumulation of DNA aberrations. Nevertheless, upregulation of mutagenic repair pathways could promote therapy resistance ultimately leading to disease progression [125]; for this reason, compounds inhibiting NHEJ or MMEJ players are in development to help in overcoming this problem and improve clinical outcomes of classic therapies.

Regarding HR, several inhibitors have been developed, targeting MRN complex and among them, mirin is the most studied (Table 2).

Mirin inhibits MRE11-associated exonuclease activity preventing MRN-dependent ATM activation. This inhibitor potentially sensitizes different cancers to genotoxic agents, such as malignant glioma, prostate cancer, and neuroblastoma [24,126,127]. In addition, several studies suggest that MRN defects could synergize with PARP inhibition, indicating a possible synthetic lethality interaction that could be exploited in specific cancer therapy [128].

As overexpression of *RAD51* stimulates cancer aggressiveness, it has been shown that lowering its expression or activity (through inhibition) can sensitize cancer cells to chemotherapeutics [113]. To date, different classes of RAD51 inhibitors which take advantage of different inhibition mechanisms have been developed (Table 2): some inhibitors interfere with RAD51 ssDNA binding ability and could lead to the formation of toxic RAD51 complexes; other inhibitors interfere with RAD51 ability to interact with other DNA repair factors such as BRCA2 and finally drugs that inhibit the RAD51 capacity to migrate through the DNA strand and form foci after DNA damage. B02 is a RAD51 inhibitor which interferes with ssDNA binding ability and has proved to sensitize cancer cells to several DNA damage agents in preclinical studies also in combination with PARP inhibitors [129,130,131]. The RAD51 inhibitor RI-1 destabilizes RAD51 oligomerization/filament formation on DNA strand leading to the formation of cytotoxic complexes. This compound specifically works on those cancer cells where *RAD51* is overexpressed, sensitizing them to chemo/radiotherapy and PARP inhibitors [132,133]. IBR120 is an inhibitor that disrupts RAD51 interaction with BRCA2 sensitizing cancer cells to IR [134,135]. Finally, CYT-0851 is a compound that reduced RAD51 migration to damaged DNA ends [125]. A clinical trial to test this inhibitor has already started, (NCT03997968) (Table 2). Anyhow, even if the other inhibitors are not in clinical trials, they proved to have great potential in pre-clinical studies.

Since BRCA1 and BRCA2 are large proteins that interact with several partners with their functional domains and act as hubs in DNA damage signaling/repair, it is difficult to design specific compounds to inhibit their activities [136]. Some BRCA1/BRCA2 inhibitors acting on the BRCT domain, which is essential for the interaction with DNA repair partners, such as RAD51, determine an increase of cell sensitivity to drugs such as PARP inhibitors [137,138]. Moreover, small molecules acting on the RING domain of BRCA1 have been developed and this could provide novel ways to inhibit HR [24].

RAD52 is another therapeutic target in breast and ovarian cancer because its depletion is synthetically lethal to cells BRCA2 or BRCA1/PALB2 deficient [139,140]. Different groups have developed several RAD52 inhibitors (F79, 6-OH-dopa, A5MP, AICAR/ZMP, D-I03, NP-004255, F779-0434) that act either by inhibiting oligomerization or blocking ssDNA binding activities of the protein, similar to some RAD51 inhibitors [141] (Table 1). Recently, to potentiate the effect of RAD52 drugs and reduce resistance, a new strategy named “dual synthetic lethality” has been proposed with the idea to combine them with PARP inhibitors, in BRCA1-deficient cancer cells [142].

Regarding NHEJ, several chemical compounds have been developed for cancer therapy, but they are all in preclinical studies. These studies are analyzing how their inhibition could influence cancer development and drugs resistance (Table 3). It has been reported that Ku70/80 inhibition sensitizes cells to radiation treatment, XLF inhibition contributes to overcome chemoresistance in colorectal cancer cells, and LIG IV silencing results in increased cellular sensitivity to chemotherapeutic alkylating agents [143,144,145].

Some studies are developing small ligands able to inhibit TP53BP1 by binding to its interacting domain and competing with regular substrates, making it unable to coordinate downstream effectors [146,147], moreover a synthetic lethal interaction between TP53BP1 loss and treatment with ATR inhibitors and cisplatin has been demonstrated [148] (Table 3).

MMEJ pathway is still far from being fully characterized and several groups are working to identify drugs able to inhibit Pol-θ activity and to exploit this pathway in cancer therapy [149]. Recently, it has been discovered that Pol-θ inhibition sensitizes cells to replication stress caused by agents such as topoisomerases poisons or ATR inhibitors, suggesting novel cancer treatments [150] (Table 3). In addition, thanks to a CRISPR genetic screen, 140 genes whose deletion is synthetically lethal with *POLQ* mutations have been discovered. Among them, several HR and NHEJ-involved genes and components of the TP53BP1 pathway have been identified. Moreover, in the same study, it has been discovered that 30% of breast cancer cases deposited in TCGA (the Cancer Genome Atlas) have alteration in at least one of these 140 genes; the proteins encoded by these genes could be potential targets to develop new strategies for cancer patients who have Pol-θ alteration [151]. Moreover, knockdown of Pol-θ in ovarian cancer cell lines deficient for FA factor FANCD2 enhances cell death [152]. Finally, the antibiotic novobicin is now under study as specific Pol-θ inhibitor, and preliminary results showed that this drug is able to kill HR-deficient tumor cells due to accumulation of toxic RAD51 foci [153] (Table 3).

## 9. Targeting DNA Helicases Increases Sensitivity to Several Treatments

BLM is one of the helicases that have an essential role in HR pathway, promoting DNA end resection, RAD51 filament and D-loop formation [154]. The inhibitor ML216 targets this helicase (Table 4), by competing for ATP binding and for BLM binding to DNA; anyhow, this compound showed poor aqueous solubility and cell permeability, therefore it will require further optimization [155].

DNA2 helicase allows cancer cells to resist DNA replication stresses induced by chemotherapy or radiotherapy, by promoting flap removal during DNA replication, DNA resection during repair, and stabilization and restart of stalled replication forks [156]. C5, a potent DNA2 inhibitor, has been identified by a high throughput screening; this inhibitor sensitizes cells to camptothecin and PARP inhibitors, making it interesting for cancer therapy [157] (Table 4).

*WRN* downregulation leads to mitotic catastrophe and cancer cell death [158]. Interestingly, evidences show that treatment with a WRN inhibitor (NSC 19630) induces apoptosis in leukemia cell lines [159]. NSC617145 is an improved derivative of the previous drug, which likely traps WRN on the DNA substrate, leading to sensitization of cancer cells to DNA-damaging agents. This drug sensitizes cells deficient of the FA and NHEJ pathways to mitomycin C, suggesting the possibility of a synthetic lethality approach [160] (Table 4). Recently, new small molecule WRN inhibitors, identified by a high throughput screen, have been shown to reduce proliferation of cancer cells [161].

These data suggest that DNA helicases are potential targets for cancer therapy to be exploited as novel synthetic lethal interactions. Several studies are now ongoing to identify new drugs that could provide alternative and useful strategies to sensitize several cancers to DNA damage agents [155,162] (Table 4).

## 10. Targeting Repair of Single Base Lesions and DNA Crosslinks Improves Efficiency of Cancer Treatments

A strong association between APE1 overexpression and cancer progression has been reported in BER, most likely due to an increased resistance to DNA damaging agents such as platinum compounds [163,164]. For this reason, several studies are investigating the possibility to identify bioactive compounds able to inhibit APE1 anti-proliferative activity and consequently sensitize cells to DNA damaging agents, such as bleomycin [165,166]. The APE1 inhibitor E3330 has been proved to reduce the migration of human breast cancer cells, and it thus might have therapeutic potential in metastatic breast cancer [167] (Table 5). Strikingly, it has been shown that treatment with PARP inhibitor olaparib reduces APE1 expression and this depletion increases sensitivity to olaparib treatment in breast cancer cells (Table 5) [168].

NER has a pivotal role in resolving helix-distortion DNA lesions; thus, it would be expected that depletion of proteins involved in this pathway would increase sensitivity to DNA damaging agents (Table 6). Most of the current data confirm this assumption, even if some contrasting evidences exist. Single nucleotide polymorphisms (SNPs) of NER genes could be associated to chemotherapy response in lung cancer patients and, in gastric cancer cells, low XPC expression together with the absence of translocation of XPA and XPD to the nucleus increase sensitivity to cisplatin [171,172]. However, in contrast with previous data, high XPC expression correlates with longer survival time in colon cancer patients sensitizing colon cancer cell lines to cisplatin and radiation [173]. Low expression of ERCC1 correlates with increased sensitivity to platinum in several tumor types and its depletion sensitizes cells to PARP inhibitors [174,175,176]. Compounds targeting ERCC1/XPF complex are being developed to sensitize cancer cells to DNA damage-based chemotherapy [177,178].

Considering the importance of FA pathway in resolving ICLs, synthetic lethal interactions with this pathway for the development of inhibitors have been explored (Table 6).

A siRNA-based screening identified ATM as synthetically lethal in Fanconi anemia deficient cell lines, indicating that this gene could be targeted concomitantly with a FA pathway inhibitor [185]. Moreover, combination of CHK1 inhibitor with FANCD2 depletion hyper-sensitizes lung cancer cells to gemcitabine [124]. Several molecules that inhibit specific components of the FA pathway have been identified acting at different levels; for instance, phenylbutyrate sensitizes head and neck cancer cells to cisplatin by reducing the expression of FANCS [179], curcumin acts on the mono-ubiquitination step of FANCD2 sensitizing ovarian and breast tumor cell lines to cisplatin and glioma cell lines to alkylating agents, also MLN4924 suppresses FANCD2 mono-ubiquitination sensitizing cells to DNA ICLs agents [180,181,182] (Table 6). The USP1/UAF1 complex has been identified as a druggable target: ML323 inhibits it, leading to sensitization of colorectal cancer, non-small cell lung cancer, and osteosarcoma cells to DNA-damaging chemotherapeutics and PARP inhibitors (Table 6); inhibiting this complex also compromises TLS due to reduced deubiquitination of PCNA, thus this drug could act on FA by simultaneously targeting two steps [183,184].

These data suggest that among intra and inter-strand DNA crosslinks-repair involved proteins, the most promising are the ones involved in FA pathway that could increase efficacy of the currently used anti-cancer treatments based on DNA crosslinker such as platinum compounds and mitomycin C.

## 11. Targeting MMR and Synthetic Lethality Interactions: Possible New Combination Therapies

Defects in MMR genes confer resistance to platinum-based chemotherapy and re-expression of these components restores sensitivity to these compounds [169]. Although not many specific drugs that inhibit MMR genes have been proposed, it has been shown that Pol-β inhibition is synthetically lethal with MSH2/MLH1 deficiencies, likely because this inhibition determines accumulation of unrepaired oxidized nucleotides, which increases the amount of mismatches. Thus, Pol-β inhibitors (NSC666715) are promising agents for treatment of MMR-deficient cancers in combination with DNA-alkylating agents such as temozolomide (Table 5) [170].

## 12. PARP1 Inhibition Gave the Most Important Results in the Synthetic Lethality Field Promoting Development of New Drugs and Treatments

PARP1 is one of the most studied DNA-repair involved proteins having a role in different DNA repair pathways such as BER, NER, NHEJ, and HR. In addition, PARP1 also regulates chromatin decondensation and cell cycle arrest [186]. Several inhibitors of this protein have been developed, because PARP1 depletion is synthetically lethal together with deficiency in BRCA1 or BRCA2 [187,188]. The synthetic lethality interaction is likely due to accumulation of DNA damage caused by the combined depletion of BER or NER and HR repair; NER/BER depletion determines SSB accumulation which eventually generates DSB after replication fork collapse. HR depletion due to BRCA1/2 mutation prevents DSB repair, leading to cancer cell death [189]. Thus, PARP inhibitors have been proved to be greatly effective in breast and ovarian cancers carrying mutations in HR-related genes. To date, four PARP inhibitors have been approved by Food and Drug Administration (FDA) for cancer therapy: olaparib, rucaparib, niraparib, talazoparib, and other inhibitors are under study [190,191]. Several clinical trials are ongoing to approve these drugs for several cancer types such as colorectal, melanoma, and prostate cancers (NCT00535353, NCT00804908, NCT03732820) and to assess efficacy of new inhibitors in combination with DNA damaging agents (NCT01127178, NCT01311713). Combinatorial treatments with these drugs and classic treatments have been explored with almost any chemotherapeutic agents and with radiotherapy [192], recently prompting the development of inhibitors labeled with therapeutic isotopes, that have proven to be particularly effective in brain tumor models [193,194]. Moreover, discovering that PARP inhibitors present antiangiogenic activity promoted development of targeted therapies combining PARP inhibition with anti-angiogenic agents to improve treatment of ovarian cancer [195,196]. As shown in this review, several synthetically lethal interactions that exploit the combination of DNA damage induced by PARP inhibitors and inhibition of several DNA-repair related proteins are under study, particularly those ones involved in HR (Figure 5).

Despite their efficacy, resistance occurrence, often caused by a recovery of BRCA-related activity, is a common problem [197]. Several studies have investigated the network of PARP1 interactors in order to identify novel targets in combination with PARP inhibition, also exploring pathways not directly related to DNA repair. The final aim of these investigations is to reduce drug resistance, improve efficacy of currently used PARP inhibitors and eventually develop new compounds. Our group has identified novel PARP1 interactors by using a yeast wide-screening assay and selected by an integrative computational analysis the most relevant interactors that have been validated in cell-based assays [198,199]. Within the field of PARP inhibitors improvement, some studies are analyzing the landscape of secondary targets of these drugs to modulate efficacy and side-effects in the clinical therapy [200,201].

## 13. Conclusions

It is now evident that targeted inhibition of DNA repair and DDC proteins has numerous applications in cancer treatment. The kinases involved in the first phases of DNA damage recognition and cell cycle arrest such as: ATM, ATR, DNA-PK, and CHK1, have been demonstrated to be efficiently targeted and several inhibitors are currently being tested. Importantly, identifying new targets involved in specific pathways and new drugs could allow the development of more personalized therapeutic approaches that may result in reducing the possibility of side-effects and the occurrence of drug resistance.

Another promising application is the combinatorial treatment which takes advantage of simultaneous inhibition of related pathways, leading to an increase of efficacy as compared to the drugs alone. Recently, there is an emerging paradigm that DNA damage, which frequently occurs in cancer cells, leads to the expression of interferon and chemokines activating immune cells in the tumor microenvironment [202]; therefore, immunotherapy combined with drugs inhibiting DNA-repair proteins is now one of the most promising therapies. Numerous ongoing clinical trials are investigating efficacy of these treatments; for instance, PARP inhibitors combined with immune inhibitors (see clinical trials NCT03834519, NCT03851614, NCT03602859, and NCT03308942). Moreover, the efficacy of PARP inhibitors in HR-deficient cancers underlines the importance of the discovery of new synthetically lethal interactions and how these could be exploited in specific cancers, hence promoting development of personalized therapies. The progression in CRISPR/Cas9-based screening has made possible systematic analysis for synthetic lethal drug targets in human cancers leading to the identification of novel genetic cancer targets and eventually the development of effective drugs specific for different pathogenic conditions [203,204].

Finally, DNA damage is repaired by a protein network that is deeply connected; uncovering these connections might generate novel targets to increase efficacy of current in use therapies and overcome resistance development.

## Figures and Tables

**Figure 1 ijms-21-06684-f001:**
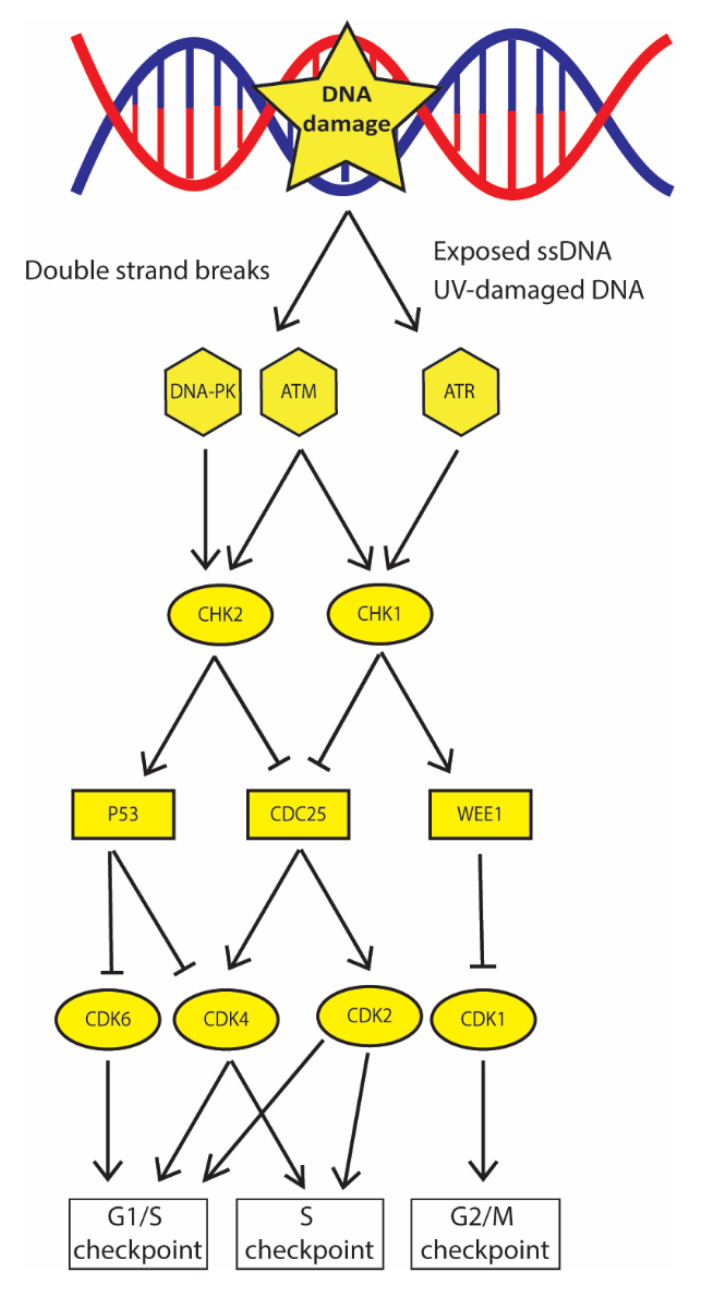
DDC signaling cascade determines arrest in specific phases of the cell cycle. ATM, DNA-PK, and ATR phosphorylate the downstream kinases CHK2 and CHK1; in turn they activate P53 and WEE1 that, by inhibiting CDK6/4 and CDK1 determine arrest in G1 and G2/S, respectively. CHK1 and CHK2 also inhibit CDC25 phosphatase that does not remove inhibitory phosphate groups from CDK2/4, therefore determining arrest in the S phase.

**Figure 2 ijms-21-06684-f002:**
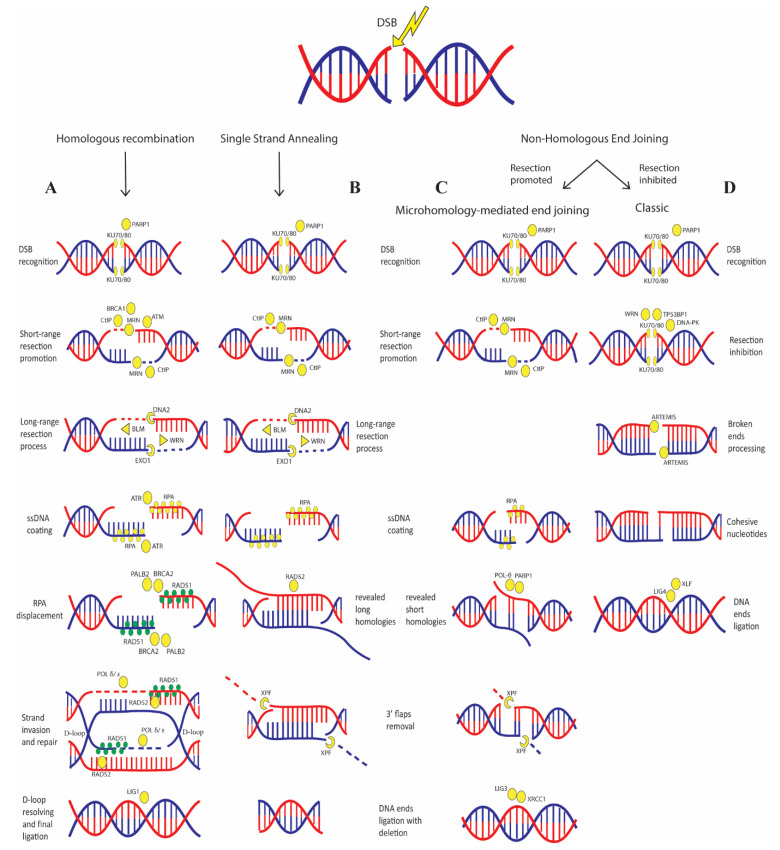
Several DSB repair pathways can be used depending on cell cycle phase and presence of homology on DNA strand. (**A**) In the S/G2 phase, when the homologous chromatid is present, HR error-free DSB repair pathway is favored: initial resection performed by the combined action of CtIP, MRN complex and BRCA1, promotes this pathway and allow access of long-range resection nucleases DNA2 and EXO1. WRN and BLM helicases unwind the two DNA strands; the ssDNA formed is rapidly stabilized by RPA and bound by ATR. Thereafter, RPA is displaced by RAD51 aided by BRCA2/PALB2 activity. RAD51 promotes search and invasion of complementary strand and RAD52 determines interaction between the complementary strands; finally, POL δ/ε re-synthetizes the damaged strand and LIG1 performs the final ligation. (**B**) Similar to HR, SSA requires long range resection of the DSB, but differently from HR this resection process exposes long regions of homology; DNA strands can bind together, also favored by the activity of RAD52, the overhanging 3′ flaps are removed by XLF nucleases and the DNA ends are ligated resulting in long deletion. (**C**) MMEJ pathway has several steps in common with SSA; the differences are that there is no long-range resection and that homology between strands are short (few base pairs). Short homology is revealed by the activity of Pol θ-associated helicase together with PARP1 and, as in SSA, the 3′flaps are removed by XLF nuclease; in the last step, XRCC1 and LIG3 catalyze DNA ends ligation resulting in short deletion. (**D**) c- NHEJ is favored by resection inhibition performed by Ku70/80 heterodimer and several other factors modulated by TP53BP1. Ku70/80 activates DNA-PK kinase which recruits ARTEMIS nucleases to the DSB; this promotes the processing of the broken ends until cohesive nucleotides are found, then XLF and LIG4 catalyze the final ligation.

**Figure 3 ijms-21-06684-f003:**
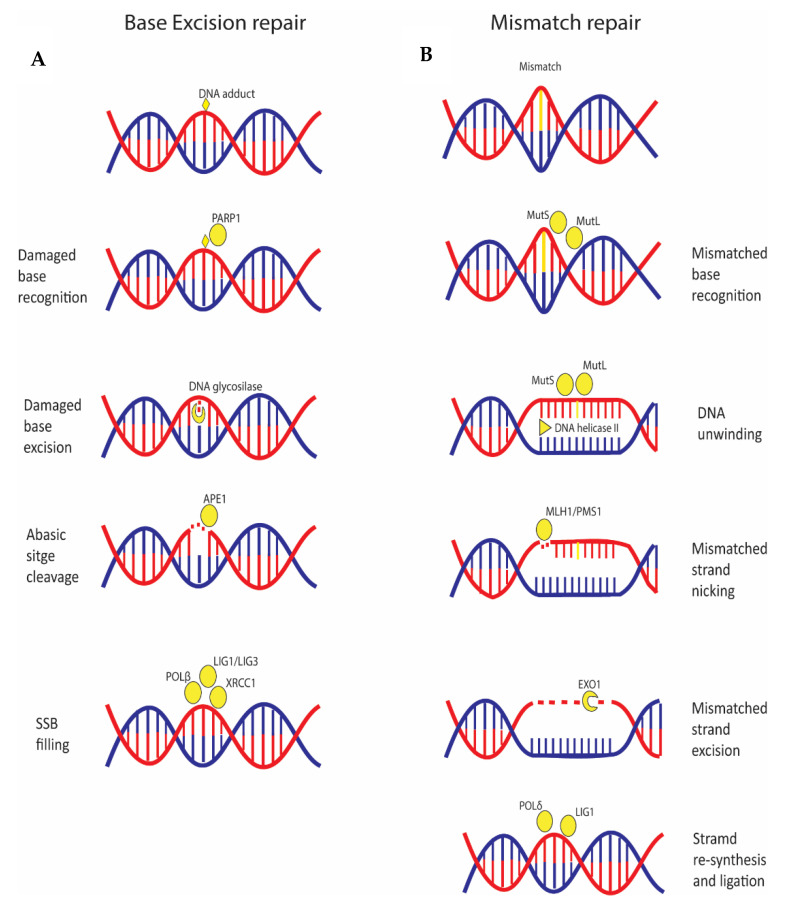
Small base lesions are repaired by base excision repair; mismatched base pairs and insertion/deletion loops are corrected by mismatch repair pathway. (**A**) DNA lesion is recognized by PARP1 which recruits a DNA glycosylase that removes the damaged base; then, APE1 digests the abasic site determining a SSB that is repaired by the combined action of LIG1/3, XRCC1 and Pol-β. (**B**) The mismatched base is recognized by the complexes MutS and MutL that, in turn, recruit the DNA helicases II to unwind the DNA strands; thereafter, the complex formed by MLH1/PMS1 digests DNA several bases from the mismatch, forming a nick. The EXO1 nuclease digests the DNA from the nick towards the mismatch; finally, the removed strand is re-synthetized by Pol-δ and ligated by LIG1.

**Figure 4 ijms-21-06684-f004:**
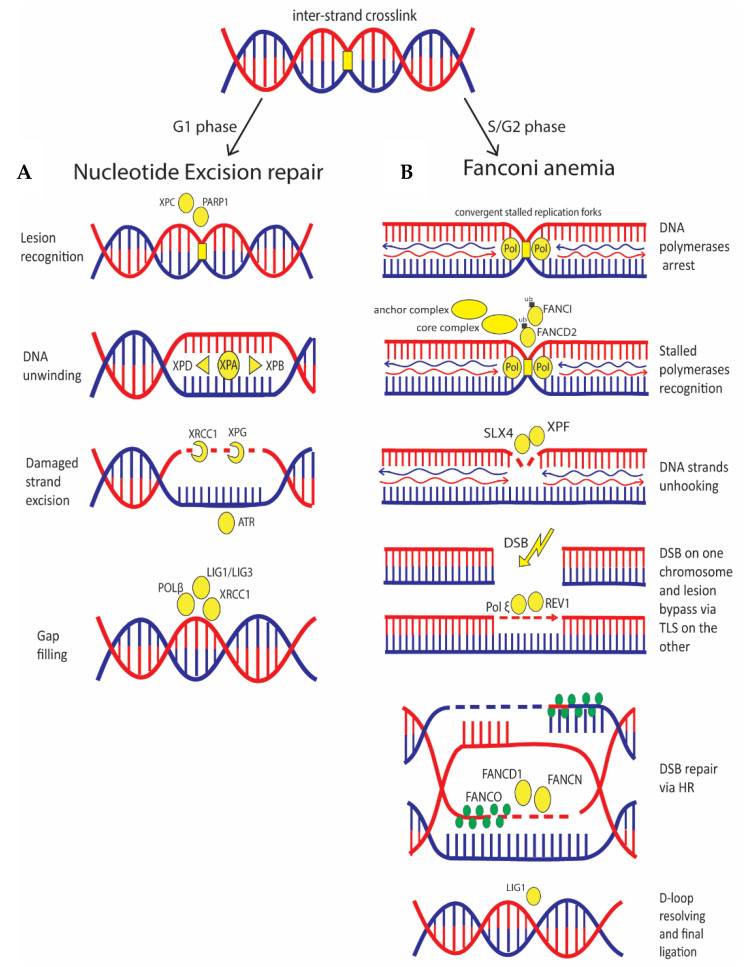
Bulky lesions are repaired by nucleotide excision repair, inter-strand crosslinks formed in S/G2 phase, are repaired by Fanconi Anemia pathway. (**A**) In NER, the bulky lesion is recognized by PARP1 and XPC; they recruit XPA that acts as a scaffold organizing the helicases XPB and XPD that unwind the DNA around the damage. Then, several bases are removed by ERCC-1-XPF and XPG endonuclease; the gap generated is filled by Pol-β and ligated by XRCC1, LIG3/LIG1. (**B**) Convergent arrested-replication forks are recognized by the anchor complex; this recruits the core complex which attaches a single ubiquitin residue to FANCI and FANCD2. After, SLX4 and XPF nucleases unhook the DNA strand determining a DSB on one strand. The strand without the DSB is repaired by TLS by Pol ξ assisted by REV1; the other strand is repaired via HR by FANCS (BRCA1), FANCD1 (BRCA2), FANCN (PALB2) and FANCO (RAD51).

**Figure 5 ijms-21-06684-f005:**
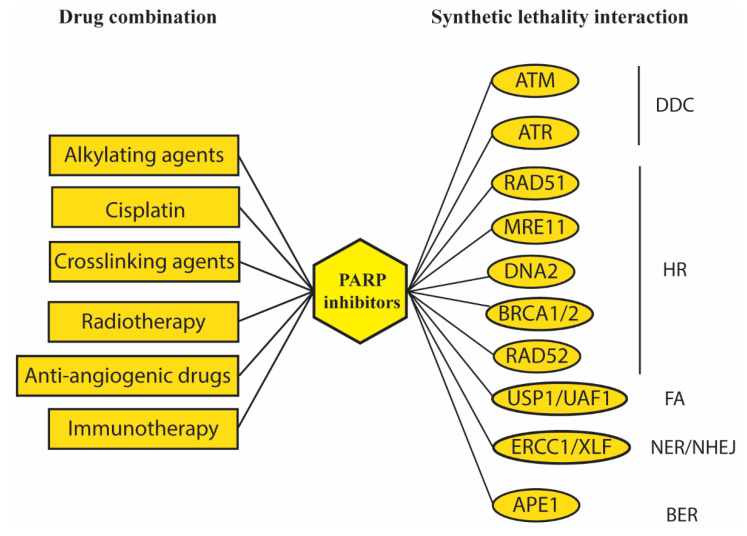
Field of PARP inhibitors in cancer therapy widely explored numerous combinations. Combination of PARP inhibitors with classic chemotherapeutic agents (alkylating agents, cisplatin, cross-linking agents), radiotherapy, targeted therapy (anti-angiogenic drugs) and immunotherapy. Numerous synthetic lethality interactions have been explored: from initial DNA damage recognition (DDC) to DNA repair, most promising results have been obtained combining PARP inhibition with DSB repair inhibition (HR, FA, NHEJ).

**Table 1 ijms-21-06684-t001:** DDC-related proteins targeted in cancer therapy.

DDC Pathway					
Protein	Drug Name	Combined Treatment	Possible Synthetic Lethal Interaction	Clinical Trial	References
ATM	AZD0156	Ionizing radiation Topoisomerase I and II inhibitors	PARP1 inhibition	NCT02588105	[105,106,107,108]
	AZD1390			NCT03423628	
	KU-55933				
	KU-60019				
	KU-59403				
ATR	M6620	Ionizing radiation Cisplatin	ATM inhibition	NCT02589522 NCT02567422	[115,116,117,118,119,120,121,122,123]
	AZD6738	Ionizing radiation	ATM inhibition WEE1 inhibition		
	BAY1895344	Ionizing radiation Cisplatin	ATM inhibition PARP1 inhibition		
	VE-821	Ionizing radiation Anti-metabolites	ATM inhibition CHK1 inhibition		
DNA-PK	M3814	Topoisomerase I and II inhibitors Anti-metabolites Ionizing radiation	ATM inhibition	NCT03770689	[109,110,111,112,113]
	VX-984	Ionizing radiation Topoisomerase I and II inhibitors		NCT02644278	
	CC-115	Ionizing radiation		NCT02516813	
	MSC2490484A	Ionizing radiation		NCT02833883	
CHK1	AZD7762	Anti-metabolites	ATR inhibition	NCT00937664	[120]
	MK-8776		FANCD2 inhibition		[124]
WEE1	AZD1775	Cisplatin Ionizing radiation	PARP1 inhibition	NCT03028766	[119]

For each target it is reported the name of the drugs currently in analysis; combination of target inhibition and classic anti-cancer therapy under study; possible synthetic lethality interaction; ongoing clinical trials with the drugs; references.

**Table 2 ijms-21-06684-t002:** HR repair proteins targeted in cancer therapy.

HR					
Protein	Drug Name	Combined Treatment	Possible Synthetic Lethal Interaction	Clinical Trial	References
MRE11	Mirin		PARP1 inhibition		[126,127,128]
RAD51	B02	Ionizing radiation Topoisomerases I and II inhibitors	PARP1 inhibition		[129,130,131,132]
	RI-1	Cisplatin Ionizing radiation			[132,133]
	IBR120	Ionizing radiation			[134,135]
	CYT-0851			NCT03997968	[125]
BRCA1			PARP1 inhibition		[24,137,138]
BRCA2					
RAD52	F79 6-OH-dopa D-I03	Cisplatin	PARP1 inhibition BRCA2 inhibition BRCA1/PALB2 inhibition		[139,140,141,142]
	A5MP AICAR/ZMP		BRCA1 inhibition		
	NP-004255 F779-0434		BRCA2 inhibition		

For each target it is reported: name of the drug currently in analysis; combination of target inhibition and classic anti-cancer therapy under study; possible synthetic lethality interaction; ongoing clinical trials with the drug; references.

**Table 3 ijms-21-06684-t003:** NHEJ repair proteins targeted in cancer therapy.

NHEJ				
Protein	Drug Name	Combined Treatment	Possible Synthetic Lethal Interaction	References
TP53BP1	i53	Cisplatin	ATR inhibition	[146,147,148]
	UNC2170			
KU70/80		Ionizing radiation		[143]
LIG4		Alkyating agents		[145]
XLF	G3	Cisplatin	PARP1 inhibition	[144]
		Anti-metabolities		
Pol-θ	Novobicin	Topoisomerase I and II inhibitors	ATR inhibition	[150]
			FANCD2 inhibition	[152,153]

For each target it is reported: name of the drugs currently in analysis; combination of target inhibition and classic anti-cancer therapy under study; possible synthetic lethality interaction; ongoing clinical trials with the drug; references.

**Table 4 ijms-21-06684-t004:** Helicases and nucleases targeted in cancer therapy.

Helicases/Nucleases				
Protein	Drug Name	Combined Treatment	Possible Synthetic Lethal Interaction	References
BLM	ML216			[155]
DNA2	C5	Topoisomerase I and II inhibitors	PARP1 inhibition	[157]
WRN	NSC 19630			[159,160,161]
	NSC617145	Mitomycin C	FANCD2/DNA-PK inhibition	

For each target it is reported: name of the drugs currently in analysis; combination of target inhibition and classic anti-cancer therapy under study; possible synthetic lethality interaction; references.

**Table 5 ijms-21-06684-t005:** BER/MMR-related proteins targeted in cancer therapy.

BER/MMR				
Protein	Drug Name	Combined Treatment	Possible Synthetic Lethal Interaction	References
APE1	E3330	Bleomycin	PARP1 inhibition	[163,164,165,166,167,168]
POL-β	NSC666715	Alkylating agents	MSH2/MLH1 inhibition	[169,170]

For each target it is reported: name of the drugs currently in analysis; combination of target inhibition and classic anti-cancer therapy under study; possible synthetic lethality interaction; references.

**Table 6 ijms-21-06684-t006:** NER/FA-related proteins targeted in cancer therapy.

NER/FA				
Protein	Drug Name	Combined Treatment	Possible Synthetic Lethal Interaction	References
ERCC1	NSC16168	Cisplatin	PARP1 inhibition	[174,175,176,177,178]
FANCS	Phenylbutyrate	Cisplatin		[179]
FANCD2	Curcumin	Cisplatin	ATM inhibition CHK1 inhibition	[124]
	MLN4924	DNA cross-linking agents		[180,181,182]
USP1/UAF1	ML323	Topoisomerase I and II inhibitors	PARP1 inhibition	[183,184]

For each target it is reported: name of the drugs currently in analysis; combination of target inhibition and classic anti-cancer therapy under study; possible synthetic lethality interaction; references.

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
