# Peer review of "Inhibition of DNA Repair in Cancer Therapy: Toward a Multi-Target Approach"

_ijms, 2020, doi:10.3390/ijms21186684_

Round 1
Reviewer 1 Report
Cancer depends on genomic instability that allows the progression through malignancy and, therefore, often display one or more impaired DNA repair pathways. On the other side, targeting DNA repair pathways is a promising strategy to obtain synthetic lethality of cancer cells.
In this review, the authors summarize the DNA repair pathways and cell phase check points, highliting those factors that are found altered in cancers. In the second part of the paper, they describe the state of the art of DNA repair targeting strategy, describing the most promising inhibitors od DNA repair factors.
The topic is of interest and the paper well summarize the topic.
Major issues:
Some parts of the work are difficult to read. In particular, I suggest reviewing sections 1, 2, and 13, if possible with the help of a proofreader with good written scientific-English skills
Minor Issues:
page 2, line 59. The phrase is difficult to understand. Please rephrase.
Page 2, line 73. “focusing and” -> “focusing on”
Page 3, lines 96-99. The sentence is difficult to understand. Please rephrase.
Page 4, lines 116-118. The sentence falls in a paragraph dedicated to mutations on ATR, ATM, and DNA-PK. So, it is not clear why the author insert here a lonely sentence regarding ATR-CHK1 axis and mitotic catastrophe. Please explain/expand/move n another paragraph.
Page 4, line 120. “After initial recognition [of DNA damage] and cell cycle arrest…
The authors may expand the BER chapter, in particular discussing the role of pol b on cancer onset. A recent paper appeared on Oncogene (https://doi.org/10.1038/s41388-020-1386-1) describing a protective role of pol beta. The authors may also consider whether mutations in pol beta linked to tumor onset may be of interest for this review (This is just an article in the topic I am aware of: 10.1021/acs.biochem.7b00869. The author should explore the literature to find eventually more appropriate papers).
Author Response
Response to reviewer 1 comments:
Major issues:
• Some parts of the work are difficult to read. In particular, I suggest reviewing sections 1, 2, and 13, if possible with the help of a proofreader with good written scientific-English skills
the manuscript was edited by an English-speaker former scientist (Michael Minks) and by Marcella Simili for scientific suggestions. The "acknowledgments" section was added to thank them and the other members of the labs who carefully read the review. Sections, 1, 2 and 13 were then re-edited.
Minor issues:
• page 2, line 59. The phrase is difficult to understand. Please rephrase.
page 2 line 59: we changed the phrase
• Page 2, line 73. “focusing and” -> “focusing on”
page 2 line 73: corrected
• Page 3, lines 96-99. The sentence is difficult to understand. Please rephrase.
page 3 line 96-99: we re-phrased and added a sentence to explain it
• Page 4, lines 116-118. The sentence falls in a paragraph dedicated to mutations on ATR, ATM, and DNA-PK. So, it is not clear why the author insert here a lonely sentence regarding ATR-CHK1 axis and mitotic catastrophe. Please explain/expand/move n another paragraph.
page 4, lines 116-118: we completely changed the sentence; now it is much clearer
• Page 4, line 120. “After initial recognition [of DNA damage] and cell cycle arrest…
page 4, lines 120: we added "of DNA damage"
The authors may expand the BER chapter, in particular discussing the role of pol b on cancer onset. A recent paper appeared on Oncogene (https://doi.org/10.1038/s41388-020-1386-1) describing a protective role of pol beta. The authors may also consider whether mutations in pol beta linked to tumor onset may be of interest for this review (This is just an article in the topic I am aware of: 10.1021/acs.biochem.7b00869. The author should explore the literature to find eventually more appropriate papers).
the ber Chapter was indeed expanded as suggested (line 292-296) and the references added
Reviewer 2 Report
In this review, the authors are summarizing most of the key pathways involved in cell cycle checkpoint and DNA repair focusing on how their alterations could predispose to cancer. The authors are also touching the inhibitors developed or in development specifically targeting different proteins involved in each pathway, underlining the rationale behind their usage and how their combination and/or exploitation as adjuvants to classic therapies could help in patient’s clinical outcome. Overall, this is a comprehensive and well-organized review on the potential application of the DNA repair pathway for cancer treatment. However, before I recommend this article to be published in IJMS, the authors should address the following points:
- Line 95-96: As the authors introduced in Section 1, together with HR and NHEJ, MMEJ is also important for DSB repair. However, the signaling pathway for MMEJ activation has not been uttered in Section 2 at all. Which kinase is responsible for the activation of the MMEJ pathway? ATM or DNA-PK? This should be discussed here.
- Line 243: There is a more recent and appropriate reference for this sentence.
- DNA oxidation and excision repair pathways (Int J Mol Sci. 2019 Dec 3;20(23):6092. doi: 10.3390/ijms20236092)
- Line 290: The more comprehensive and appropriate reference for this sentence is the following one.
- Transcriptional and Posttranslational Regulation of Nucleotide Excision Repair: The Guardian of the Genome against Ultraviolet Radiation (Int J Mol Sci. 2016 Nov 4;17(11):1840. doi: 10.3390/ijms17111840)
- Line 325: It is hard to say that all cancer cells are deficient in DNA checkpoint and repair pathway. Therefore, this sentence should be toned down and needs to be rewritten as follows. Cancer cells deficient in DNA damage checkpoint and repair pathways are particularly sensitive to DNA damage ~~.
Author Response
Response to reviewer 2 comments
1. Line 95-96: As the authors introduced in Section 1, together with HR and NHEJ, MMEJ is also important for DSB repair. However, the signaling pathway for MMEJ activation has not been uttered in Section 2 at all. Which kinase is responsible for the activation of the MMEJ pathway? ATM or DNA-PK? This should be discussed here.
Point 1 line 95-96: we added a sentence to explain how kinases are recruited to DNA damage that generate ssDNA and a sentence to explain which kinase is responsible for MMEJ activation (line 210-211)
2. Line 243: There is a more recent and appropriate reference for this sentence.
3. DNA oxidation and excision repair pathways (Int J Mol Sci. 2019 Dec 3;20(23):6092. doi: 10.3390/ijms20236092)
4. Line 290: The more comprehensive and appropriate reference for this sentence is the following one.
5. Transcriptional and Posttranslational Regulation of Nucleotide Excision Repair: The Guardian of the Genome against Ultraviolet Radiation (Int J Mol Sci. 2016 Nov 4;17(11):1840. doi: 10.3390/ijms17111840)
Point 2-5: all the references suggested were added to the text and in the reference list
6. Line 325: It is hard to say that all cancer cells are deficient in DNA checkpoint and repair pathway. Therefore, this sentence should be toned down and needs to be rewritten as follows. Cancer cells deficient in DNA damage checkpoint and repair pathways are particularly sensitive to DNA damage ~~.
Point 6: the sentence was changed as suggested
Round 2
Reviewer 1 Report
The authors have responded to all my comments. I do not recognize any further issues.